

**Title:** Mapping the Vegetation of Lake Tana Basin in Ethiopia Based on Google Earth
Images
**Running title:** Vegetation Map of Lake Tana Basin
**Authors:** Song Chuangye[a], Lisanework Nigatu[b], Yibrah Beneye[c], Abdurezak
Abdulahi[b], Zhang Lin[a] , Wu Dongxiu[a]
**Email of Authors:**
Song chuangye: songcy@ibcas.ac.cn
Lisanework Nigatu: lisaneworkn@yahoo.com
Yibrah Beneye: yibrah_beyene@yahoo.com
Abdurezak Abdulahi: arezak6@gmail.com
Zhang Lin: zhanglin@ibcas.ac.cn
Wu Dongxiu: wudx@ibcas.ac.cn
**Author's institutional affiliations:** [a]State Key Laboratory of Vegetation and
Environmental Change, Institute of Botany, Chinese Academy of Sciences, Beijing,
China; [b]School of Natural Resource and Environmental Sciences, Haramaya
University, Dire Dawa, Ethiopia; [c]College of Plant and Horticultural Sciences,
Hawasa University, Hawasa, Ethiopia

**Abstracts** Lake Tana basin is one of the most important watersheds of Nile basin. It is
of great importance for the economy and politics of Ethiopia. In past decades, natural
vegetation of Lake Tana basin was heavily destroyed for continuous expansion of
cropland. Vegetation conservation and restoration have to be performed in order to
protect natural environment and maintain the biodiversity in Lake Tana basin. To
provide detailed information of actual vegetation for planning of vegetation
conservation and restoration, in this research we mapped the vegetation of Lake Tana
basin based on high spatial resolution images provided by Google earth and field
survey data, through the approach of visual interpretation. A total of 31972 polygons
were generated to represent the vegetation patches of Lake Tana basin on the map,
and the validation based on surveyed vegetation plots indicated that 90.6 % patches



were correctly identified. The statistics of vegetation map indicated that natural
vegetation (natural forest, woodland, bush land, grassland and wetland) occupies
14.32 % of the basin area, plantation forest occupies 1.92 % of the basin area,
cultivated land occupies 61.8 % of the basin area, water body, village and urban
occupy 20.8 %, 0.68 % and 0.46 % of the total area of Lake Tana basin respectively.
Doi of the dataset used for map production is
http://doi.org/10.4121/uuid:48d45053-36f6-411b-96b1-7ae0e22d56d0. We expected
that this vegetation map could benefit vegetation conservation and restoration in Lake
Tana basin.
**Key Words** East Africa; Blue Nile; The Abbay River; Nile basin; Land cover; Visual
Interpretation

## 1 Introduction

Lake Tana, located in highlands of North-West Ethiopia, is the largest fresh water
lake of Ethiopia, and the third largest lake of Nile Basin. Lake Tana is the source of
Blue Nile with the basin being one of the most important catchments of Nile Basin. It
has rich natural resources and great potential for the development of irrigation,
hydroelectric power, high value crops, aquatic products, livestock products and
ecological tourism (Bijan and Shimelis, 2011). Lake Tana basin is of critical national
significance in economy and politics of Ethiopia. It also has great influences on
livelihoods of tens of millions of people in lower Nile basin.
Historically, large area of afromontane forest and many indigenous plant species
existed in Lake Tana basin. 172 woody species were observed in Lake Tana basin, and
many of them were indigenous species (IFAD, 2007a). There are also large areas of
wetlands and seasonally flooded plains in Lake Tana basin. They are the source of
multiple services for local community and the home of many endemic bird species
(Ayalew, 2010; Bijan and Shimelis, 2011).
The population density of Lake Tana basin is very heavy and the rate of
population growth is very high. More than two million people reside in this basin, and



the population density is greater than 150 per square kilometer (Yimenu, 2005).The
great population and high rate of population growth lead to the increase of food
demand. Large area of forest, grassland and wetland were destroyed and transformed
into cropland, and more livestock were raised on the grassland. Deforestation and
overgrazing resulted in massive destruction of natural vegetation, decline in
biodiversity and forest stand density, desertification and soil erosion (Alelign et al.,
2007). In order to protect natural environment and maintain biodiversity, vegetation
restoration and conservation have to be performed in Lake Tana basin (Bishaw, 2001).
Since 1990s, many conservation efforts were undertaken to conserve and restore the
natural vegetation of Lake Tana basin (Bishaw, 2001; Teketay, 2001). However,
degradation and decline of natural vegetation in Lake Tana basin is still a major
problem (IFAD, 2007b).

Detailed data of regional vegetation distribution is the base for vegetation

management and conservation. Only when vegetation of the whole basin was well
surveyed and mapped, rational and scientific planning of vegetation conservation and
restoration could be made for the whole basin. However, vegetation maps related to
Lake Tana basin were almost made for Africa, East Africa and Ethiopia with small
scales, such as the vegetation map of Eritrea, Ethiopia and Somalia with the scale of
1:5000000 (Pichi Sermolli, 1957), vegetation map of Ethiopia and Eritrea
(Breitenbach, 1963), vegetation map of Africa with the scale of 1:5000000 (White
1983), vegetation map of Africa Horn (Friis, 1992), vegetation map of Ethiopia
(Sebsebe, 1996; Sebsebe, 2004; Sebsebe and Friis, 2009), potential vegetation map of
Ethiopia with the scale of 1:2000000 (Friis et al., 2011).However, vegetation maps
compiled by Pichi Sermolli (1957), Breitenbach (1963), White (1983) and Friis (1992)
were published many years ago and were all with small scales. They cannot provide
detailed information of actual vegetation of Lake Tana basin. Potential vegetation map
compiled by Friis et al. (2011) could not reflect the status of actual vegetation of Lake
Tana basin either. Another map concerned to the vegetation of Lake Tana basin is that
the land cover/use map made by Shimelis et al. (2008) with the scale around
1:1700000. However, only large patches of vegetation were mapped, and many



patches of vegetation were merged or omitted on this map. Therefore, shortage of
detailed vegetation data in Lake Tana basin limited the effectiveness of planning of
vegetation management and biodiversity conservation. Therefore, in this research,
based on high spatial resolution satellite images provided by Google earth and field
survey data, we made a vegetation map of Lake Tana basin. We hope this map will be
helpful for the vegetation and biodiversity conservation in Lake Tana basin.
**2 Study Area**
Lake Tana is located on highlands of North-West Ethiopia (Figure 1). The
average altitude of Lake Tana is around 1800 meters. The area of Lake Tana basin
(including water surface area) is 15096 km$^2$. The water surface area is 3000-3600 km$^2$
and the maximum depth of water is 14 meters. Gilgel Abay, Ribb, Gumera and
Megech are the most important rivers feeding Lake Tana and contribute more than 90%
of total inflow.
The zonal vegetation of Lake Tana basin is dry evergreen afromontane forest.
However, only small patches of remnant forest exist currently due to heavy
deforestation. The biodiversity of Lake Tana basin is rich and many endemic plant
species grow in this catchment. There are large areas of wetlands in this basin. These
wetlands are the home of many endemic birds.

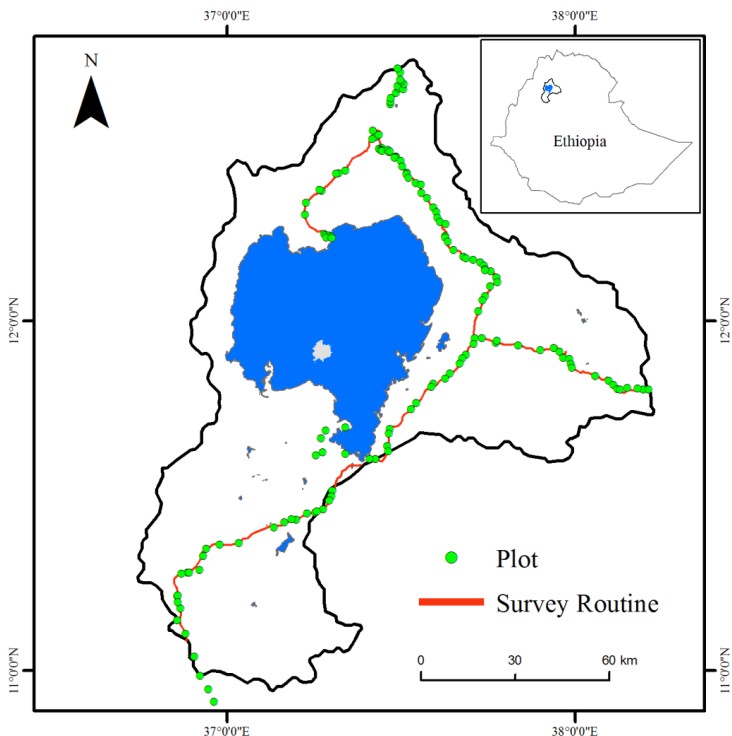

Figure 1 The location of Lake Tana basin, the survey route and plots

## 3 Data and Method

### 3.1 Data Sources

Vegetation mapping was based on high spatial resolution satellite images and aerial images provided by Google earth and collected vegetation survey data. Field vegetation surveys were performed in 2015, 2016. Total 156 vegetation plots were investigated (Figure 1) and dominant species were recorded in the field. In addition to this, "Atlas of the Potential Vegetation of Ethiopia" compiled by Friis et al. (2011) was also important references in this research.

### 3.2 Vegetation Classification System

Based on vegetation classification system adopted by Shimelis et al. (2008) and suggestions from geobotanists of Ethiopia, vegetation of Lake Tana basin was categorized into seven groups: natural forest, woodland, plantation forest, bushland, grassland, wetland, and cultivated land. Three types of non-vegetation cover, waterbody, village and urban, were also mapped in this research. There are sub-types



of these vegetation groups exist for variation of dominant species. However, we did
not differentiate these sub-types for the limitation of spatial resolution of satellite
images.

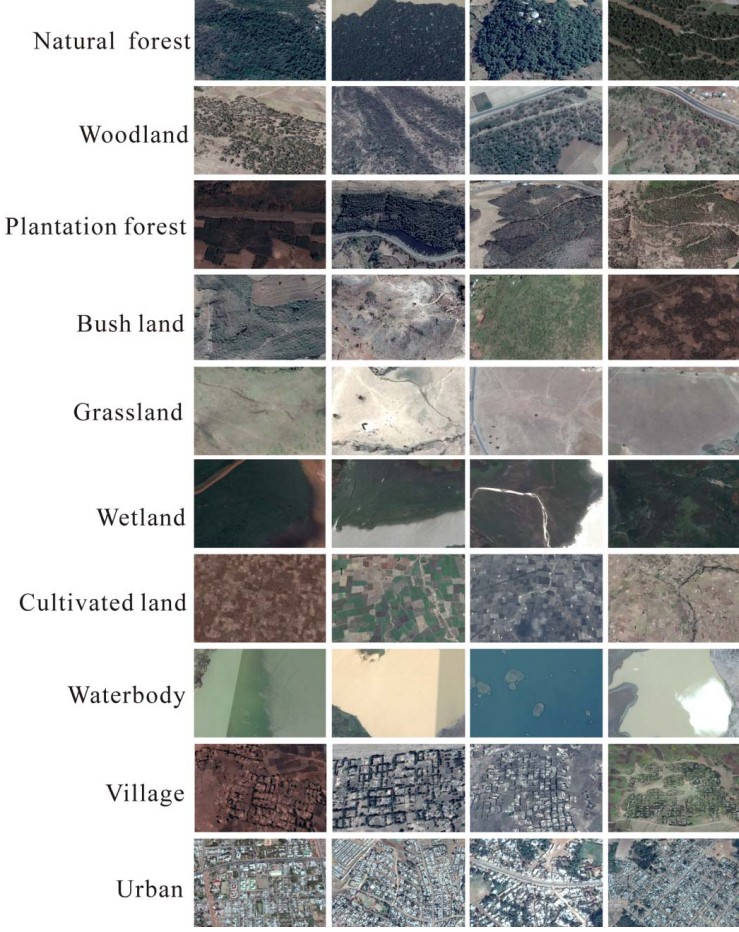


Figure 2 Interpretation marks based on the Google earth images

**3.3 Method**

Coordinates of vegetation plots were recorded and then transformed into kml

files, which could be read by Google earth.

One-third of surveyed plots were randomly selected to establish interpretation

marks. Open these kml files in Google earth, and established interpretation marks
according to characteristics of color and texture of vegetation reflected on satellite
images (Figure 2).The other two-third surveyed plots were used to validate



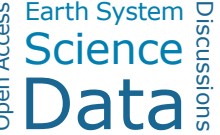

interpretation results.

On Google earth, visual interpretation was employed to identify vegetation based

on established interpretation marks. The tool "Add polygon" was used to vectorise
vegetation patches around the scale of 1:5000. The whole process lasted more than
one and a half year, and 31972 polygons were generated to represent vegetation
patches of Lake Tana basin on the map.

In order to improve the accuracy of vegetation interpretation, advices and

suggestions of Ethiopian geobotanist were often consulted to determine vegetation
type. The validation based on surveyed vegetation plots indicated that 90.6 % patches
were correctly identified.

Kml files of all vegetation were imported into Global Mapper software (v16.0),

and then transformed into shp files which could be read by ArcGIS (v9.3, ESRI). In
ArcGIS, vegetation type of each polygon was marked in attributes table and all shp
files were merged into one shp file. Finally, vegetation map was designed and
exported for printing on A1 (at the scale about 1:310000) (Figure 3).
**3.4 Projected Coordinate System and Geographic Coordinate System**

Projected Coordinate System:WGS_1984_UTM_Zone_37N; Projection:

Transverse_Mercator; False_Easting:500000.00000000; False_Northing:0.00000000;
Central_Meridian:39.00000000; Scale_Factor:0.99960000;
Latitude_Of_Origin:0.00000000; Linear Unit: Meter.

Geographic Coordinate System: GCS_WGS_1984; Datum: D_WGS_1984;

Prime Meridian: Greenwich; Angular Unit: Degree.
**4 Results**
**4.1 Natural Forest**

1320 patches of natural forest were identified and vectorised, and the total area is

122.2 km$^2$, which occupy 0.82% of the area of Lake Tana basin. The area of
maximum and minimum patch is 12.6 km$^2$ and 0.00075 km$^2$, respectively. The mean
area of natural forest patches is 0.093 km$^2$.

Two types of natural forest exist in this basin: dry evergreen afromontane forest



and riverine forest (Friis et al., 2011). The altitude where dry evergreen afromontane
forests occur ranges from 1500 m to 2700 m. The mean annual temperature is
14-25 °C and the mean rainfall is 700-1100 mm (Friis, 1992). High amplitude of
altitude and rainfall result in complex habitats and species composition. Characteristic
species of arborous layer are *Podocarpus falcatus* and *Juniperus procera*. Dominant
species of understory are *Croton macrostachyus*, *Ficus* spp., *Oleaeuropaea* subsp.
cuspidata, *Trema orientalisand* and *Maesa lanceolata*.

Riverine forest is predominantly located near lake and river. Dominant species

are *Diospyros mespiliformis*, *Mimusops kummel* and *Syzygium guineense*.

Due to continual expansion of cropland, natural forest was gradually destroyed in

past decades. Only small patches of remnant forests can be found in two main forms
in this region: protected state forests and church forest.
**4.2 Woodland**

1613 patches of woodland were identified and vectorised, and the total area is

236.1 km$^2$, which occupy 1.58 % of the area of Lake Tana basin. The area of
maximum and minimum patch is 5.6 km$^2$ and 0.0023 km$^2$ respectively. The mean area
of woodland patches is 0.15 km$^2$.

There are two kinds of woodland in Lake Tana basin: *Combretum-Terminalia*

woodland and *Acacia-Commiphora* woodland (IBC, 2005; Friis et al., 2011).

*Combretum-Terminalia* woodlands occupy the area with altitude of 500–1900 m.

They are usually located in humid areas of lowlands or on valley of rivers.
Characteristic species of *Combretum-Terminalia* woodland are *Combretum* spp.,
*Terminalia* spp., *Oxytenanthera abyssinica*, *Boswellia papyrifera*, *Anogeissus*
*lieocarpa*, *Sterospermem kuntianum*, *Pterocarpus lucens*, *Lonchocarpus laxiflorus*,
*Lannea* spp., *Albizia malacophylla* and *Enatada africana*. Most of them are small
trees with large deciduous leaves. They often grow together with *Oxytenanthera*
*abyssinica*. The understory is a mixture of herbs and grasses. Dominant herbal species
include *Justecia* spp., *Barleria* spp., *Eulophia* spp., *chlorophytum* spp., *Hossolunda*
*opposita* and *Ledeburia* spp..

*Acacia-Commiphora* woodlands usually occupy dry slope with the altitude of



1000-1900 m (WBISPP, 2004). Habitats are characterized with quite large variations
of soil and topography and diverse biotic and ecological elements. Most of these plant
species in *Acacia-Commiphora* woodland haves mall deciduous leaves or leathery
evergreen leaves.
There is a large variation of stand density for *Acacia-Commiphora* woodlands.
*Acacia-Commiphora* woodlands could be observed with three kinds of formation:
dense forest with close canopy, scattered individuals, even wooded grassland.
*Acacia-Commiphora* woodlands are also famous for some *Acacica*, *Boswellia* and
*Commiphora* species. They could be used to produce gum and resin.

**4.3 Plantation Forest**

11390 patches of plantation forest were identified and vectorised, and the total
area is 287.1 km$^2$, which occupy 1.92 % of the area of Lake Tana basin. The area of
maximum and minimum patch is 1.73 km$^2$ and 0.00064 km$^2$ respectively. The mean
area of plantation forest patches is 0.025 km$^2$.
*Eucalyptus* species are the main species of plantation forest. *Cupressus lusitanica*
and Pine species were also planted in some areas. In addition to this, *Acacia mearnsii*
was also found to be planted in the southern area of Lake Tana basin.
There are around 600 *Eucalyptus* species in the world and more than 120 species
were found in Ethiopia (Alemayehu, 2017). *Eucalyptus globuls* and *Eucalyptus*
*camaldulensis* are the most common and widely planted species in Ethiopia.
*Eucalyptus globulus* was usually planted in the area with altitude over 2200 m, and
*Eucalyptus camaldlunesis* was planted in the region with altitude of 1700-2400 m.
The plantation of *Eucalyptus* species was widely criticized from the suppression
effects of growth of associated indigenous species and heavy use of underground
water. However, plantation area of *Eucalyptus* forest increased rapidly in past fifteen
years (Birru et al., 2003).

**4.4 Bushland**

12023 patches of bushland were identified and vectorised, and the total area is
792.3 km$^2$, which occupy 5.3 % of the area of Lake Tana basin. The area of maximum
and minimum patch is 16.9 km$^2$ and 0.0004 km$^2$ respectively. The mean area of

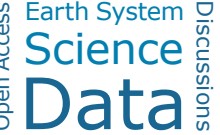

bushland patch is 0.066 km$^2$.

Bushland often occurs in the area with shallow soil and steep slope, such as hills,

escarpments, mountains and gorge slopes. There is usually grassland on the bottom of
bushland. This forms bush-grass complex. The dominant woody species of bushland
are *Maytenus senegalensis*, *Carissa spinarum*, *Clausene anista*, *Clerodendrum*
*myricoides*, *Grewia ferruginea*, *Caesalpinia decapetala*, *Ficus verruculosa*,
*Calpurnia aurea*, *Erica arborea*, *Hypericum rebolutum*, *Vernonia* spp., *Senna* spp.,
*Cordia* spp., *Acacia* spp., *Commiphora Africana* and *Indigofera* spp..

**233    4.5 Grassland**

4083 patches of grassland were identified and vectorised, and the total area is

595.8 km$^2$, which occupy 3.99 % of the area of Lake Tana basin. The area of
maximum and minimum patch is 7.29 km$^2$ and 0.0016 km$^2$ respectively. The mean
area of grassland patches is 0.15 km$^2$.

Grasslands mainly distribute along rivers, around villages, on mountain and hill

tops, on slopes and on highlands with stony and shallow soils. Common species are
*Eragrostis* spp., *Pennisetum* spp., *Panicum* spp., *Echinochloa* spp., *Setaria* spp.,
*Hyparrhenia* spp., *Cymbopogon* spp., and *Sorghum* spp.. Scattered shrubs could be
observed on the grassland, such as *Senna* spp., *Maytenus senegalensis*.

**243    4.6 Wetland**

1030 patches of wetland were identified and vectorised, and the total area is

393.5 km$^2$, which occupy 2.63 % of the area of Lake Tana basin. The area of
maximum and minimum patch is 9.41 km$^2$ and 0.0015 km$^2$ respectively. The mean
area of wetland patches is 0.38 km$^2$.

Wetlands are distributed around the Lake and along tributaries of the lake.

*Hygrophila auriculata*, *Cyprus papyrus*, *Typha latifolia*, *Phragmites australis*,
*Nymphaea caerulea*, *Juncus dregeanus*, *Floscopa glomerata*, *Eriocaulon* spp., *Xyris*
*capensis* are the main species of wetlands.

Wetlands have rich biodiversity and diverse ecological functions. The lake and

its tributaries are the home of 28 fish species, of which15 are endemic species to
Ethiopia. More than 300 species of birds have been observed and recorded in Lake



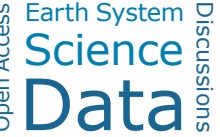

Tana basin, which was defined as an international bird site by BirdLife International
(BLI) (Shimelis, 2013).

### 4.7 Cultivated Land

The area of cultivated land is 9239.6 km$^2$, which occupies 61.8% of the total area
of Lake Tana basin. Teff, sorgum, chickpea, rice, maize and sesame are widely planted
in Lake Tana basin. These crops are often planted mixed with endemic or exotic arbor
species, such as *Croton macrostachyus*, several *Acacia* species, *Albizia gummifera*,
*Cordia africana*, *Juniperus procera*, *Grevillea robusta* and *Sesbania sesban*, which
formed complex agroforestry system.
Many kinds of fruits are planted in agroforestry, like *Mangifera indica*, *Persea*
*americana*, *Carica papaya*, *Citrus sinensis*, *Citrus aurantifolia*, *Rhamnus prinoides*,
*Mimusops kummel*and *Syzygium guineense*.

### 4.8 Waterbody

37 patches of waterbody were identified and vectorised, and the total area is
3112.4 km$^2$, which occupy 20.8 % of the area of Lake Tana basin. The area of
maximum and minimum patch is 3080.8 km$^2$ and 0.0017 km$^2$ respectively. The mean
area of waterbody patch is 84.1 km$^2$.
Lake Tana is the biggest waterbody in this watershed. The total area of Lake
Tana is 3080.8 km$^2$, which occupy 98.98% of total water surface area.

### 4.9 Village

476 patches of village were identified and vectorised, and the total area is 100.97
km$^2$, which occupy 0.68 % of the area of Lake Tana basin. The area of maximum and
minimum patch is 2.24 km$^2$ and 0.002 km$^2$ respectively. The mean area of village
patch is 0.21 km$^2$.
In Lake Tana basin, the size of many villages is very small. These small villages
distribute sparsely in the landscape. It is difficult to vectorise all the village patches.
Therefore, only large villages were identified and vectorised in this research.

### 4.10 Urban

There are two big cities in Lake Tana basin: Gondar and Bahir Dar. The total area
of urban is 69.04 km$^2$, which occupy 0.46 % of Lake Tana basin.



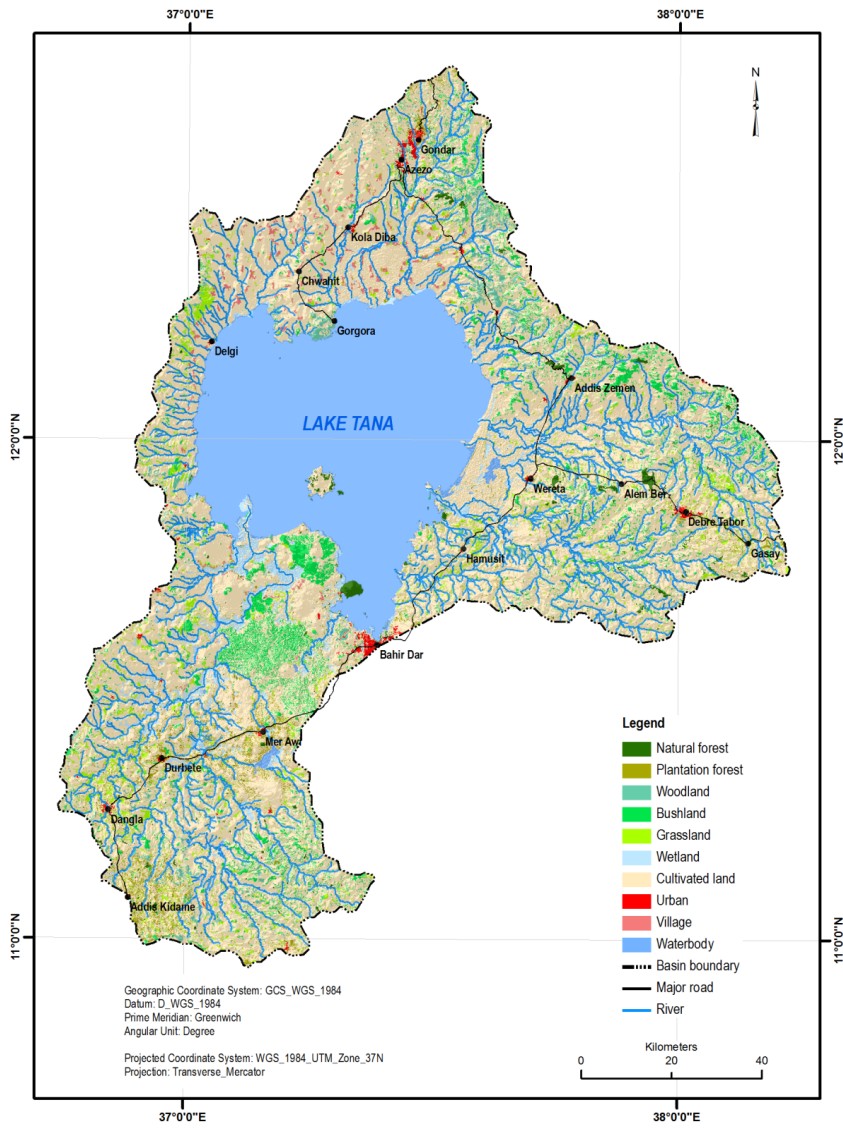


Figure 3 Vegetation map of Lake Tana basin, Ethiopia

**5 Discussions and Conclusions**

Satellite images and aerial images provided by Google earth offered us valuable

and free information for vegetation mapping. The high spatial resolution makes it
possible for us to identify small patches of vegetation by visual interpretation. In this
research, the validation indicated that most of vegetation patches were correctly
identified. We believe this vegetation map could offer reliable information for



vegetation conservation in Lake Tana basin.

The potential vegetation of Lake Tana basin is dry evergreen afromontane forest

and grassland complex (Friis et al., 2011), which should cover most area of this basin.
However, based on this vegetation map, we found that natural vegetation only
occupies 14.32 % of the basin area. Among natural vegetation, the percentage of
natural forest is below 40 %, and bush lands, most of them being secondary
vegetation, share 37 % of the total area of natural vegetation. This reflected the
seriousness of vegetation degradation in Lake Tana basin.

The number of natural vegetation patches is as high as 20069 with the total area

of 2140 km$^2$. The mean patch area of natural forest, woodland, bush land, grassland
and wetland is only 0.07km$^2$, 0.08 km$^2$, 0.60km$^2$, 0.20 km$^2$and 0.05 km$^2$, respectively.
This indicated that natural vegetation was highly fragmented.

Vegetation degradation and fragmentation reduced the geographical environment

for the survival of biological species and influenced the flow of material and energy
balance in the ecosystem. These will definitely impact the maintenance of biodiversity.
In addition to this, vegetation degradation also results in soil erosion and
desertification. Therefore, more and larger conservation areas are needed to maintain
the biodiversity and protect the environment in Lake Tana basin.

Plantation forest (dominated by *Eucalyptus* species) occupies 1.9 % of the area

of Lake Tana basin. But, plantation of *Eucalyptus* was proved to have negative
influences on the maintenance of biodiversity and ecological water balance, for its
allelopathy and high consumption of water (Martens, 2002; Cornish and Vertessy,
2001). Moreover, we need to realize that allelopathic effect depends on the amount of
rain falling in the site and texture of the soil. Not all *Eucalyptus* species release the
same concentration of allelochemicals (Pohjonen and Pukkala, 1990; Lisanework and
Michelsen, 1993, 1994; Michelsen et al., 1996). Therefore, site-species matching and
objectives of plantations should clearly be defined to avoid negative connotations
against *Eucalyptus*. These actions will maximize the economic benefits and minimize
the ecological risk brought by *Eucalyptus*.

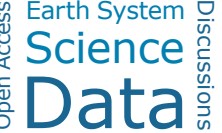


**Data Access**:http://doi.org/10.4121/uuid:48d45053-36f6-411b-96b1-7ae0e22d56d0.

**Author contributions:** Wu Dongxiu and Song Chuangye designed the research, Song Chuangye, Lisanework Nigatu, Yibrah Beneye, Abdurezak Abdulahi, Zhang Lina, Wu Dongxiu collected the data, and Song Chuangye wrote the manuscript. Wu Dongxiu and Lisanework Nigatu revised the manuscript.

**Competing interests:** The authors declare that they have no conflict of interest.

**Acknowledgments:** This study was funded by the Ministry of Sciences and Technology (International Science & Technology Cooperation Program of China 2014DFG32090) and State Key Laboratory of Vegetation and Environmental Change, Institute of Botany, Chinese Academy of Sciences.

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
