# Peer review of "Title: Mapping the Vegetation of Lake Tana Basin in Ethiopia Based on Google Earth"

_Earth System Science Data, 2018_

## Referee Comment (RC1) · Anonymous Referee #1 · 1 Jun 2018

**Comments on "Mapping the Vegetation of Lake Tana Basin in Ethiopia Based on Google Earth Images"**

This MS introduces a vegetation dataset of Lake Tana Basin for potential uses in vegetation management and conservation. Vegetation types were delineated using the method of human visual interpretation. The accuracy of the dataset was validated by ground survey. The authors have put great labor efforts to delineate such a detailed vegetation map. It is a valuable dataset to document vegetation in this specific area. It may be too much to ask the authors to improve the quality of the dataset at this stage, but more details on methodology and discussions on accuracy of the dataset should be provided for future uses.

**Manuscript**
1. This is a new dataset that has potential uses in landscape management and land process modeling in the basin of Lake Tana, Ethiopia.
2. The method of producing the dataset is not new, but valid.
3. The mothed is not described in sufficient detail.
   3.1. Even though the paper cites Shimelis et al. (2008) for the vegetation classification system. I suggest describe this important information for the convenient of data use.
   3.2. L132: Number of the plots should be clearly provided, even though you have shown in Figure 1 in the MS.
   3.3. L144: Validation accuracy matrix should be provided (accuracy of each land use/cover class), not just an overall accuracy (90.6%). The data quality has not been fully assessed regarding different vegetation classes.
4. Potential error sources should be evaluated and discussed, such as errors in visual interpretation: How many people? Who were they? Were they from local areas? How did they work (time, place, machines, etc.)? How were they trained to do the work? How did they consult to the geobotanist in Ethiopia (L143)?
5. The manuscript should focus on methodology and data accuracy. The Results, Discussions, and Conclusions belong to interpretation of data, which is outside the scope of ESSD articles (https://www.earth-system-science-data.net/about/aims_and_scope.html). However, the possible uses of the dataset should be discussed.

**Data quality**
1. The dataset is accessible.
2. Readme file (meta data)
   2.1. "In the attributes table of this °∞VegCover°± dataset…": unexpected text codes (it could be my computer problem).
   2.2. "Because, except the cover type 1-9, lake tana basin was mostly under cultivation": grammar: "except for", "types", "Lake Tana Basin". Please check the rest of the file.
   2.3. Suitable software for visualization of the dataset should be pointed out as required by the journal.
3. Cultivated lands were not identified in the dataset. That is not consistent with the manuscript (Figs. 2 & 3 in the MS). Without identification of cultivated lands, any statement

in Section "4.7 Cultivated Land" is not accurate. This also limited the use of dataset in land use/cover change: human disturbance indicated by cultivated lands.

4. I understand that it is labor-intensive for visual interpretation. Most boundary delineation is OK. But some polygon boundaries do not have the accuracy that is claimed in L139: polygons were added in the scale of 1:5000 (Fig 1 here).  It should be revealed and discussed.

5. There are big or small gaps between polygons. They can be cultivated lands that were not identified, or gaps that should be merged to neighbor polygons that have the same attributes (Fig 2 here). It is not usual leaving gaps in maps of land use/cover.

6. It is hard for me to visually identify 2-plantation forest from 3-natural forest (Fig 2 here). Figure 2 in the MS is helpful, but more detailed description about the difference between the classes and identification criteria is recommended for the sake of accuracy assessment and future uses. The gaps are not necessarily cultivated lands, implied by the "readme" file.

7. There might be a mismatch between the time of Google map images (undocumented in the MS, meta data show 2016-2017) and validation time (i.e., 2015 and 2016, L144). My random samples indicate some misclassifications of 2-plantation forests, which look like cultivated lands (Fig 4 here). The misclassifications may be due to land use changes and the time difference of the basemap I used here and the basemap in Google Map that was used for visual interpretation for this dataset. Please document the years of images that were used for this dataset. Please discuss the validation issue related to the mismatch of time. Even though it is costly practical using Google Images, it is not the best practice using unknown (time) images to for interpretation of land use/cover.

[Figure]

Fig 1. Screenshot of polygon boundary on ArcMap 1:5000 scale. Basemap is ArcGIS online base images (imported on Jun 1, 2018, same for Fig 2, 3, and 4 here).

[Figure]

Fig 2. Gaps and classification accuracy. Map is on ArcMap 1:5000 scale. Arrows from the top to bottom indicate: the 1st and 2nd: unnecessary gaps between class 7 (wetlands); the 3rd, miss classification between 3 (natural forest) and 7; the 4th, unidentified island; the 5th, boundary error.

[Figure]

Fig 3. The visual difference between 2-plantation forest and 3-natural forest is very small. Map is on ArcMap 1:5000 scale.

[Figure]

Fig 4. Some cultivated lands were classified as 2-plantation forests. ArcMap 1:5000 scale.

---

## Author Comment (AC1) · 5 Jul 2018

This MS introduces a vegetation dataset of Lake Tana Basin for potential uses in vegetation management and conservation. Vegetation types were delineated using the method of human visual interpretation. The accuracy of the dataset was validated by ground survey. The authors have put great labor efforts to delineate such a detailed vegetation map. It is a valuable dataset to document vegetation in this specific area. It may be too much to ask the authors to improve the quality of the dataset at this stage, but more details on methodology and discussions on accuracy of the dataset should be provided for future uses.

RESPONSE Dear Professor: Thank you very much for your comments and sugges-

tions on this manuscript and the dataset. Your comments and suggestions are of great importance for the improvement of the quality of the manuscript and the dataset. We have made the following revisions according to your comments and suggestions:(1) introduction about the vegetation classification system of Shimelis et al. (2008) was added to the manuscript (section 3.2);(2) Keys to image interpretation were summarized in section 3.3; (3) the detailed information about the way how we consulted to Ethiopian geobotanists was added to this manuscript (section 3.4);(4) the accuracy matrix of ground validation was added to this manuscript (section 4.1); (5) in discussion part, more details on the accuracy of the dataset, vectorization and potential uses of this dataset were added to this manuscript; (6) we revised the dataset and corrected some misclassification, re-edited some polygons to make the boundaries more fit to the vegetation patch. We aslo added 746 new polygons to this dataset, most them are villages, grasslands and bushlands. However, limited by the time, I am afraid that not all the flaws existed in this dataset were thoroughly eliminated.

Best regards Song (On behalf of all the authors)

Manuscript 1. This is a new dataset that has potential uses in landscape management and land process modeling in the basin of Lake Tana, Ethiopia.

RESPONSE Thank you for your positive comments on this dataset. We hope this dataset could benefit the management of local forestry, agriculture and land resources.

2. The method of producing the dataset is not new, but valid.

RESPONSE The vectorization was made by the tool "add polygon" provided by google earth software.The ideneification of vegetation was performed by human visual interpretation. This method is easy and convinent to be used by the people, and the economic cost is low, but labor intensive and time consuming.

3. The method is not described in sufficient detail. 3.1. Even though the paper cites Shimelis et al. (2008) for the vegetation classificationsystem. I suggest describe this

important information for the convenient of data use.

RESPONSE Description on the vegetation classification system of Shimelis et al (2008) was added to section 3.2. Among the systme of Shimelis et al., some vegetation types are very difficult to be differentiated from each other, such as forest-mixed and forest-evergreen, pasture and range-grasses.So,based on the suggestions of Ethiopian geobotanists, we merged these similar types in this research.

3.2. L132: Number of the plots should be clearly provided, even though you have shown inFigure 1 in the MS.

RESPONSE The information about the total number of plots, number of plots used for interpretation marks, and the number of plots used for validation was added to section 3.1 and 3.3.

3.3. L144: Validation accuracy matrix should be provided (accuracy of each land use/coverclass), not just an overall accuracy (90.6%). The data quality has not been fully assessedregarding different vegetation classes.

RESPONSE We performed accuracy assessment based on the surveyed plots. And the accuracy matrix (Table 1) was added to the manuscript in section 4.1. In this matrix, for each vegetation type, the percent of correctly identified was shown in this matrix. And also, the percent of not correctly identified was also presented in this matrix. Taking bushland for example, we presented the percent of correct identification (85%), we also caculated the percent of bushland plots classified as other vegeattion types (15%, this means that 15% of surveyed plots of bushland were identified as woodland).

4. Potential error sources should be evaluated and discussed, such as errors in visual interpretation: How many people? Who were they? Were they from local areas? How didthey work (time, place, machines, etc.)? How were they trained to do the work? How didthey consult to the geobotanist in Ethiopia (L143)?

RESPONSE The vectorization was done by three persons. They are Chinese technicians who are familiar with Google earth software. Before the vectorization, a training course was held to help them to learn the detailed requirements of vectorization, such as the scale, the boundary and characteristics of each vegetation type. The vegetation identificationwas made by only one person. This aims to keep the consistentence of the identification criterion. We consulted to Ethiopian geobotanists in two ways: face to face co-working and via Email. More information about the introduction of visual interpretation has been added to section 3.3.

5. The manuscript should focus on methodology and data accuracy. The Results, Discussions,and Conclusions belong to interpretation of data, which is outside the scope of ESSD articles(https://www.earth-system-science-data.net/about/aims_and_scope.html). However, thepossible uses of the dataset should be discussed.

RESPONSE Yes, the manuscript should focus on the methodology and data accuracy. Therefore, in method section, We added more information about the construction of interpretation marks, vectorization and keys of image interpretation. In the result section: data accuracy matrix was generated based on the surveyed plots (Table 1). The percent of correctly identified and uncorrectly identified were both presented in this matrix. About the description of vegetation/land cover, this information might be usefule for readers to learn more information about the vegetation/land cover, such as area, structure, species composition, habitat characteristics and so on. Therefore, description of vegetation and land cover were kept in this manuscript. In discussion section, most of the original discussions were deleted. we analysized the error sources of vegetation identification and problems existed in the vectorization. And we also discussed the potential uses of this dataset.

Data quality 1. The dataset is accessible.

RESPONSEïïjŽThe dataset has been uploaded to the 4TU.ResearchData. Everybody

could download the data from http://doi.org/10.4121/uuid:48d45053-36f6-411b-96b1-7ae0e22d56d0.

2. Readme file (meta data) 2.1. "In the attributes table of this °∞VegCover°±dataset…": unexpected text codes (itcould be my computer problem).

RESPONSE I did not find this phenomenon on my computer. I donot know why this happened. This might be the problem of the version of software.

2.2. "Because, except the cover type 1-9, lake tana basin was mostly under cultivation":grammar: "except for", "types", "Lake Tana Basin". Please check the rest of the file.

RESPONSE Thank you very much for your careful revision on the text. I have revised the text in the readme file.

2.3. Suitable software for visualization of the dataset should be pointed out as required bythe journal.

RESPONSE The message about the suitable software for visualization of these five datasets was added to the readme file.

3. Cultivated lands were not identified in the dataset. That is not consistent with the manuscript (Figs. 2 & 3 in the MS). Without identification of cultivated lands, any statementin Section "4.7 Cultivated Land" is not accurate. This also limited the use of dataset in landuse/cover change: human disturbance indicated by cultivated lands.

RESPONSE From the google images, it is easy to see that large area of Lake Tana basin is covered by cultivated lands except for these identified land covers (bushland, plantation forest, natural forest, urban, village, waterbody, wetland, woodland, grassland). We thought that the vector data of cultivated lands could be achieved by erasing these identified polygons from the vector data of the the whole basin. We admitted that some other landcovers,such as barren lands, was mixed into the catalog of cultivated land. But, the area of barren lands is small and they are not easy to be identified

through visual interpretation. So, the influences of the missing of barren lands on the spatial pattern of land cover is limited.

3. I understand that it is labor-intensive for visual interpretation. Most boundary delineation isOK. But some polygon boundaries do not have the accuracy that is claimed in L139:polygons were added in the scale of 1:5000 (Fig 1 here). It should be revealed anddiscussed.

RESPONSE Thank you very much for your understanding on the issues of boundary delineation. Although we asked the technicians to make the vectorization at the scale of 1:5000 and delineate the boundaries strictly along the margin of land covers,issues of boundaries still happened. We have revised the boundaries you put forward in Fig 1. We also checked the polygons and re-edited some polygons to make the boundaries more fit to the land cover patches. But, limited by the time, I am afraid that not all flaws existed in this dataset were thoroughly eliminated.We also discussed this issue in "Discussion" part.

5. There are big or small gaps between polygons. They can be cultivated lands that were notidentified, or gaps that should be merged to neighbor polygons that have the sameattributes (Fig 2 here). It is not usual leaving gaps in maps of land use/cover.

RESPONSE Thank you very much for your careful review on this dataset. We used the tool of "add polygon" to vectorize the vegetation/land patches. Gaps will occure if two vegetation/land patches are connected or they are very close to each other. These gaps could be cultivated lands or other kinds of land covers. However, in this dataset, these gaps were classified into cultivated lands. These gaps definitely affected the quality of this dataset. Considering that the area of gaps is not big, we think these gaps will not exert significant influences on the pattern of vegetation/land cover at the scale of the whole basin. We have re-edited the polygons to decrease the area of gaps based on your comments (Fig 2).We have also discussed this issue in discussion part of this manuscript. People who use this dataset should know this.

6. It is hard for me to visually identify 2-plantation forest from 3-natural forest (Fig 2 here).Figure 2 in the MS is helpful, but more detailed description about the difference betweenthe classes and identification criteria is recommended for the sake of accuracy assessmentand future uses. The gaps are not necessarily cultivated lands, implied by the "readme" file.

RESPONSE The canopy color of the plantation forest is homogenous. Whereas the canopy color of natural forest is more rich than that of plantation forest. The canopy texture of plantation forest is different from that of natural forest. Large crowned trees could be observed from the top of natural forest canopy. Whereas for plantation forest, the size of crown is small and uniform.In addition to this, the borders of plantation forest patches are usually straight. And the border of natural forest patches are usually curve. Although the differences between natural forest and plantation forest is siginificant, misclassification still occurred in the process of interpretation. For the interpretation marks, we added more words to describe the characteristics of these vegetation types and keys of image identification (section 3.3). About the gaps, most of these gaps are cultivated lands. And some gaps are barren lands or other land covers. But, on the whole basin, the area of barren lands and other land covers is small, compared with cultivated lands.

7. There might be a mismatch between the time of Google map images (undocumented in theMS, meta data show 2016-2017) and validation time (i.e., 2015 and 2016, L144). Myrandom samples indicate some misclassifications of 2-plantation forests, which look likecultivated lands (Fig 4 here). The misclassifications may be due to land use changes and thetime difference of the basemap I used here and the basemap in Google Map that was usedfor visual interpretation for this dataset. Please document the years of images that wereused for this dataset. Please discuss the validation issue related to the mismatch of time.Even though it is costly practical using Google Images, it is not the best practice usingunknown (time) images to for interpretation of land use/cover.
RESPONSE The visual interpretation was performed in 2016 and 2017. And the imaging time of Google earth photos used forinterpretation were mostly 2015, 2016 or 2017. Unfortunately, we did not record the exact imaging time of the photos used for vegetation interpretation. The plots were collected in 2015 and 2016. Therefore, there exist mismatches between imaging time of Google earth photos and validation time. This will lead to some uncertainties for the validation.We also discussed this issue in section 5.1. Plantation forests are mostly eucalyptus forests. The eucalyptus will be cut for charcoal production after 3-5 years of growth. Therefore, the plantation forests may look like cultivated lands after the trees were cut. The advantage of mapping vegetation by Google earth image is money saving, but really time consuming and labor intensive.

Please also note the supplement to this comment:
https://www.earth-syst-sci-data-discuss.net/essd-2018-14/essd-2018-14-AC1-supplement.zip

---

## Referee Comment (RC2) · Anonymous Referee #2 · 20 Aug 2018

Review ESSD 2018-14, high res imagery of Lake Tana basin

Many errors in text, authors need systematic reading and correction.

Page 5 line 113 "aerial" Over most of the land masses of the planet Google Earth does NOT ingest aerial (e.g. airborne) but only satellite sources. If the authors know of specific aerial images included in specific Google Earth products, they should identify those sources and those images. Otherwise this statement seems inaccurate.

Page 5 line 118, classification system. Why did the authors not use IGBP land use categories? Their list here does not match other classification systems and therefore does not allow external comparisons. If they have valid scientific reason to keep these categories, they should show how their categories cross-match to IGBP categories.

One assumes - for lack of detailed information from the manuscript - that these authors have used very high resolution Google Earth images, e. g. at or better than 5 m resolution. But: a) most Google Earth images do not include proper metadata, e.g. source files (LandSat, SPOT, etc.), processing techniques, exact times, actual resolution, etc.; and b) recently Google has started to mix in older LandSat images - of lower resolution - in their time sequences at many locations. These authors do not assure us which images they used and at which resolution. They should look at Pinzon and Tucker 2014 (Remote Sensing 2014, 6(8), 6929-6960; doi:10.3390/rs6086929), and particularly at panel E of Figure 4 in that manuscript. What, if anything, have they provided here that improves on those prior AVHRR-based NDVI data? I agree that they might have a better product but they have failed to demonstrate how or on what basis.

The manuscript lacks a serious discussion of methods, validation and uncertainties. It does not provide guidance or assurance for other users. The entire Results section relies mostly on previous surveys with little reference to this work. The authors have largely avoided and ignored the largest land category, cultivated lands.

In addition to identification errors as pointed out by the other reviewer, this reader expects two other sources of error: changes in the satellite sources applied image by image by Google Earth (which the authors have not mentioned and perhaps can not in fact extract from the GE images themselves); and survey / validation uncertainty. Other on-the-ground land use surveys involving human observers (e.g. see the Woods et al. survey of landscape features in the UK, https://doi.org/10.5194/essd-10-899-2018) describe in great detail observer training, paper-based and digital support tools, digitising and annotation errors and exceptions, subsequent replication by experts, etc. We get no idea from this manuscript how or based on what criteria they involved their expert(s). If the authors do not understand and address uncertainties, readers / users get no basis on which to build trust in these polygons and categories.

Most of the results section seems drawn from prior literature rather than from analysis of these data. Given errors, uncertainties, and the decision to avoid cultivated lands, this reader doubts that the authors can justify and document discussion in four significant figures (e.g. natural forest at 0.82% or plantation forest at 1.92%). In places these authors invoke temporal changes in land use but they provide absolutely no data here to document those changes.

The data downloads cleanly from 4TU but the .txt file only recites the categories without providing additional useful information or metadata, at least some of the files for the individual categories open only in ESRI ArcGIS, and users get no access to summary information they would need to check the authors' results.

Given substantial absences of information in this manuscript and data as presented, this reviewer does not agree that (line 292) "this vegetation map could offer reliable information for vegetation conservation in Lake Tana basin".